# The Role of the Lymphocyte Transformation Test in Immune-Related Adverse Events from Immune Checkpoint Inhibitors: A Case Series

**DOI:** 10.3390/jcm14238596

**Published:** 2025-12-04

**Authors:** Fiorela C. Dueñas Lopez, Zoraida del Solar Moreno, Daniela Aguilar-Concepción, Carmen Ruiz-Fernández, Ibtissam Akatbach-Bousaid, Olga Rogozina, Susana Martín-López, Ana Martínez Feito, Miguel González-Muñoz, Elena Ramírez

**Affiliations:** 1Clinical Pharmacology Department, Faculty of Medicine, La Paz University Hospital-IdiPAZ, Autonomous University of Madrid, 28046 Madrid, Spain; fiorelacinthia.duenas@salud.madrid.org (F.C.D.L.); zoraida.delsolar@salud.madrid.org (Z.d.S.M.); olga.rogozina@salud.madrid.org (O.R.); smartinlopez@salud.madrid.org (S.M.-L.); 2Immunology Department, La Paz University Hospital-IdiPAZ, 28046 Madrid, Spain; daniela.aguilar@salud.madrid.org (D.A.-C.); carmen.ruiz@idipaz.es (C.R.-F.); ibtissam.karemes@salud.madrid.org (I.A.-B.); amartinezf@salud.madrid.org (A.M.F.)

**Keywords:** immune checkpoint inhibitors (ICIs), immune-related adverse events (irAEs), lymphocyte transformation test (LTT), causality assessment, rechallenge, case series, pharmacovigilance, oncology

## Abstract

**Background**: Immune-related adverse events (irAEs) represent a considerable complication associated with the use of immune checkpoint inhibitors (ICIs) in oncology patients. Assessing causality is particularly challenging in patients administered multiple therapeutic agents. The Lymphocyte Transformation Test (LTT) may aid in causality analysis, although its clinical utility remains investigational. **Objectives**: To evaluate the potential utility of the Lymphocyte Transformation Test (LTT) in the causality analysis of irAEs in cancer patients and to assist clinicians in the decision-making process regarding ICI rechallenge. **Methodology**: We present a case series of seventeen cancer patients who developed irAEs during ICI therapy. Causality was assessed using the Spanish Pharmacovigilance System, and LTT was performed for both ICIs and co-medications. **Results**: Events primarily affect hepatic, respiratory, and renal functions. Five patients showed a positive LTT to ICIs (range 3.9–13.6), all with high-grade irAEs, and none underwent rechallenge. Six patients were positive for concomitant drugs: three tested positive for both. The initial high-grade irAEs rate was 53.9%. After rechallenging, 81.8% had all-grade and 45.5% had high-grade irAEs, predominantly pneumonitis. **Conclusions**: LTT may provide supportive evidence in complex irAE cases and assist clinicians in decision-making regarding ICI rechallenge. More prospective studies are needed.

## 1. Introduction

Immune checkpoint inhibitors (ICIs), including agents targeting cytotoxic T-lymphocyte-associated antigen 4 (CTLA-4), programmed cell death protein 1 (PD-1), and its ligand PD-L1, have become the standard of care treatments across a broad spectrum of malignancies. Notwithstanding their clinical efficacy, ICIs are associated with a unique array of toxicities, which are often unpredictable and can be severe, collectively referred to as immune-related adverse events (irAEs). These adverse events are classified by severity, ranging from grade 1 (mild) to grade 5 (death), in accordance with the Common Terminology Criteria for Adverse Events (CTCAE) (Haanen et al., 2022) [1]. IrAEs are frequently caused by autoimmune mechanisms initiated when ICIs interfere with immune self-tolerance, resulting in inflammatory damage across various organ systems, occasionally with potentially life-threatening outcomes.

High-grade irAEs (grades 3–4) require prompt recognition and intervention, typically involving the discontinuation of ICI therapy and the initiation of systemic corticosteroids or other immunosuppressive agents. While many irAEs resolve with appropriate management, the decision to rechallenge patients with ICIs after a severe event remains complex and clinically nuanced.

Recent evidence suggests that ICI rechallenge can be both safe and effective in carefully selected patients. Comparative data indicate comparable objective response rates (ORR) between initial and rechallenged ICIs; however, the risk of irAE recurrence remains. Meticulous patient selection and vigilant monitoring are crucial for mitigating risks and enhancing outcomes (Fujita et al., 2018; Nomura et al., 2017; Ravi et al., 2020; Santini et al., 2018) [2,3,4,5]. The mean event rate for overall irAEs at any grade was 40.0% (37.3–42.7%), with high-grade events occurring at 19.7% (15.8–23.7%). The mean event frequencies for ICI monotherapy, ICI combination therapy, and the use of both ICI monotherapy and combination therapy were 30.5% (28.1–32.9%), 45.7% (29.6–61.7%), and 30.0% (25.3–34.6%), respectively (Jayathilaka et al., 2025) [6].

In clinical practice, patients receiving ICIs are often prescribed concomitant medications to manage comorbid conditions. The simultaneous pharmacological activity of these agents can complicate the identification of the true causative agent in the event of an adverse reaction, as their pharmacodynamic effects interact within the same biological system. Therefore, a comprehensive causality assessment is imperative. This process typically involves a detailed clinical history, targeted laboratory investigations, and the use of validated algorithms to assess causality. In this study, causality was assessed using the drug causality algorithm of the Spanish Pharmacovigilance System. When deemed necessary, additional diagnostic tools—such as the Lymphocyte Transformation Test (LTT)—were employed to further clarify the probable source of irAEs.

The LTT, an in vitro assay that evaluates T-cell activation in response to specific drugs, may offer supportive information in distinguishing between irAEs attributable to ICIs and those triggered by concomitant medications. Although its use in oncology remains investigational, the LTT may contribute to pharmacovigilance efforts and individualized decision-making, particularly when considering whether to safely rechallenge immunotherapy after an adverse event.

Given the clinical complexity and individualized nature of irAE management, particularly in the context of ICI rechallenge, there is a pressing need for real-world data to inform decision-making. While randomized controlled trials offer high-level evidence, they often exclude patients with prior severe irAEs or multiple comorbidities, limiting their generalizability. In contrast, case series offer valuable insights into clinical heterogeneity, therapeutic decision-making processes, and outcomes in real-life settings, where standard protocols may not be fully applicable. They enable the systematic description of patient characteristics, causality assessments, and treatment outcomes, facilitating the identification of patterns that may inform future clinical practice.

Therefore, the objective of this study was to describe the clinical characteristics and outcomes of a case series of oncology patients who developed irAEs during ICI treatment. Specific objectives include evaluating the contribution of the LTT to the causality assessment of these events, particularly in differentiating the involvement of the ICI from that of concomitant medications, and analyzing how this information influenced the clinical decision to rechallenge with immunotherapy, as well as describing the outcomes of the patients who were rechallenged.

## 2. Methodology

A retrospective case series study was conducted at La Paz University Hospital in Madrid, Spain, a tertiary-care teaching facility. Since 2007, all admissions have been monitored by the Proactive Pharmacovigilance Program from Laboratory Signals in Hospital to proactively detect serious ADRs (Ramirez et al., 2010) [7].

Cases were identified through two sources: (i) routine surveillance within the institutional pharmacovigilance program; (ii) inter-specialty consultations managed by the Pharmacovigilance Unit of the Clinical Pharmacology Department (Figure 1). The study was approved by the La Paz University Hospital Ethics Committee (Code PI-3226; 25 May 2018). While informed consent for the retrospective data review was waived, specific informed consent was obtained from all patients who underwent flow cytometry (results published in Ruiz-Fernández et al., 2025) [8] and participated in inter-specialty consultations at the hospital.

Each case underwent a structured causality assessment using the Spanish Pharmacovigilance System (SPS) algorithm, complemented by LTT to evaluate T-cell sensitization. Clinical outcomes following ICI rechallenge were recorded, and descriptive analyses were performed to summarize demographic, clinical, and immunological data.

The inclusion criteria were patients who were treated with ICIs for various malignancies and who subsequently developed an irAE requiring clinical intervention. For all patients with suspected adverse drug reactions, a complete report was submitted to the pharmacovigilance center in Madrid, Spain (https://www.notificaram.es, accessed on 30 June 2025).

Data were collected from electronic health records, encompassing patient demographics, oncological history, details of ICI treatment (including drug, dose, duration), and information on all concomitant medications. For each irAE, we documented the type, time to onset, and severity, graded according to the Common Terminology Criteria for Adverse Events (CTCAE) version 5.0. A comprehensive causality assessment was performed for each case, which included the application of the Spanish Pharmacovigilance System’s (SPS) drug causality algorithm (Aguirre & García, 2016) [9]. The SPS algorithm evaluates the relationship between a suspected drug and an ADR based on seven criteria: (a) Time sequence (chronology between the start of treatment with the suspected drug(s) and the appearance of the adverse effects); (b) Identification of plausible adverse drug reactions using knowledge extracted from the literature; (c) Withdrawal effect: evolution of the adverse effect after withdrawal of the suspected medication; (d) Re-exposure effect: reaction after readministration (rechallenge) of the suspected drug; (e) Alternative explanation for the observed effects; (f) Contributing factors favoring the causal relationship (e.g., renal failure and relative overdose of a drug with predominantly renal elimination); g) Complementary explorations: serum drug levels, biopsies, positive radiological examinations, positive specific skin tests, etc. The maximum possible score is 12.

Based on the obtained scores, the causal relationship is classified as: definite (≥8), probable (6–7), possible (4–5), conditional (1–3), or unrelated (<0). An SPS score ≥ +4 was considered drug-related.

In addition, peripheral blood mononuclear cells were collected from patients to perform an in vitro LTT to assess T-cell sensitization to the implicated ICI and to any suspected concomitant drugs including polysorbates as excipients.

Lymphocyte proliferation was measured as previously described (Pichler & Tilch, 2004) [10]; see Figure 2. Briefly, mononuclear cells were isolated from fresh peripheral blood using density gradient medium (Histopaque-1077, Sigma-Aldrich, Darmstadt, Germany) and subsequently plated in flat-bottom wells of microtiter plates at 2 × 10^5^ cells per well. These cells were incubated for a period of six days with various drug concentrations in triplicate. The drugs were tested at concentrations of 1, 10 and 100 μg/mL, with occasional use of lower or higher concentrations (0.1, 200 or 500 μg), as previously described (Pichler & Tilch, 2004) [10]. In the case of the ICIs, the concentrations were 1, 10, 50, and 100 μg/mL for ipilimumab, nivolumab, and pembrolizumab, and 10, 100, 250, and 500 μg/mL for atezolizumab, durvalumab, and tremelimumab. Phytohemagglutinin (5 μg/mL) was used as a positive control. For the final 18 h of the incubation, cell proliferation was assessed by the addition of 1 μCi (3H) thymidine. Proliferative responses were expressed as the stimulation index (SI), calculated as the ratio between the mean counts per minute in cultures with the drug and those without the drug. The concentration range used in the LTT included the clinically relevant plasma levels of each drug, ensuring that the in vitro stimulation reflected systemic exposures achievable in treated patients. An LTT result was considered positive if the SI was ≥2 at any tested drug concentration, as this threshold is the most widely accepted in clinical and research practice. Five healthy controls were tested with the ICIs, and the stimulation index obtained was <2. To improve interpretability, we applied the binary classification positive/negative across all cases. In some cases, drug-induced CD69 expression on T lymphocytes was analyzed by flow cytometry as previously described (Ruiz-Fernández et al., 2025) [8]. Finally, for patients who were rechallenged with an ICI, we systematically recorded the clinical outcomes, including the incidence, type, and grade of any subsequent irAE.

The statistical analysis conducted for this study was purely descriptive. Categorical variables, such as patient sex, cancer type, and the frequency of specific irAEs, were summarized using absolute numbers and percentages. Continuous variables, such as patient age, were described using the mean and range. For ordinal data, such as the scores obtained from the SPS drug causality algorithm, the median and range were used. No inferential statistical tests were performed due to the descriptive design and the limited sample size of the case series. All data were compiled and analyzed using Microsoft Excel for Mac Version 16.103.2 (25112216) standard spreadsheet software.

## 3. Results

Seventeen cases fulfilled the inclusion criteria of having received ICIs for various malignancies and subsequently developing an immune-related adverse event (irAE) requiring clinical intervention.

### 3.1. Baseline Characteristics

Patient demographics, oncological diagnoses, and characteristics of the initial irAEs are summarized in Table 1.

The case series consisted of twelve males (70.6%) and five females (29.4%), with a mean age of 64.8 years (range: 45–76). The most common malignancy was lung cancer, identified in 8 patients (47.1%), followed by melanoma in 3 patients (17.6%) and urothelial carcinoma in 2 patients (11.8%). Other cancer types included kidney, squamous cell, cecum, and thyroid carcinoma, each in one patient. The ICIs administered were primarily PD-1 inhibitors like pembrolizumab and nivolumab, PD-L1 inhibitors such as durvalumab and atezolizumab, and the CTLA-4 inhibitors ipilimumab and tremelimumab, either as monotherapy or in combination.

The irAEs observed were diverse, affecting multiple organ systems. The most frequent events were pneumonitis, hepatitis, acute tubular interstitial nephritis (ATIN), and colitis. The severity of the initial irAEs varied, with 53.9% of patients experiencing high-grade (Grade 3–4) events. The latency from ICI initiation to irAE onset ranged from 5 to 497 days.

### 3.2. Structured Causality Assessment (SPS Algorithm)

Each case underwent a structured causality assessment using the Spanish Pharmacovigilance System (SPS) algorithm. Median SPS scores were similar for ICIs and concomitant medications (median 6 for both), indicating that clinical assessment alone was insufficient to distinguish the causative agent when multiple therapies were involved. The SPS algorithm classified most cases as either “probable” or “possible”.

### 3.3. Lymphocyte Transformation Test (LTT) and Immunological Assessment

Peripheral blood mononuclear cells were collected to perform an LTT in all cases. LTT results, also summarized in Table 2, provided additional immunological evidence supporting causality. Five patients (29.4%) had a positive LTT for an ICI (stimulation index [SI] 3.9–13.6), all of whom had experienced high-grade irAEs. Six patients (35.3%) had positive LTT responses to a concomitant medication, and three patients were positive for both an ICI and another drug. In select cases, flow cytometric assessment of CD69 upregulation supported sensitization to excipients such as polysorbate 80.

### 3.4. ICI Rechallenge and Outcomes

The decision to reintroduce an ICI following the resolution of the initial adverse event was made for eight patients. Table 3 provides a detailed account of these rechallenge cases, specifying the ICI used, the time to onset for the subsequent irAE, its type, and its grade. Eight patients who were rechallenged all experienced a recurrent or new irAE. Notably, four of these patients (50%) developed a high-grade irAE upon rechallenge. For instance, patient IRAE02, who initially presented with Grade 3 pneumonitis, experienced a recurrence of high-grade pneumonitis just 17 days after the first rechallenge with pembrolizumab and again 5 days after a second rechallenge. In contrast, patients who initially had low-grade irAEs, such as IRAE03 and IRAE05, developed different, low-grade events (arthralgias) upon rechallenge.

.

### 3.5. Integrated Interpretation of Causality and Clinical Outcomes

A comprehensive causality assessment was performed for all patients. This included applying the SPS algorithm and the LTT for both the suspected ICI and concomitant medications. The detailed results of this assessment, including LTT values, SPS scores, clinical recommendations, and patient outcomes, are presented in Table 2. The SPS algorithm yielded a probable or possible causal relationship in most cases. Five patients (29.4%) had a positive LTT result for an ICI (stimulation index range: 3.9–13.6), all of whom had experienced high-grade irAEs. Six patients (35.3%) showed a positive LTT for a concomitant medication, and three patients tested positive for both an ICI and a co-administered drug.

Final outcomes varied across the cohort, as shown in Table 2. Five patients exhibited markedly positive LTTs to ICIs (range: 3.9 to 13.6), all of whom experienced high-grade irAEs and were not rechallenged. Six patients demonstrated positive results for concomitant medications (range: 2.2–25.8), with three patients testing positive for both the ICI and the concomitant medication. In several instances, sensitization to excipients (e.g., polysorbate 80) was suspected and subsequently confirmed through positive cytometry.

The results of the causality assessment and the rechallenge outcomes are summarized in Table 4. The analysis reveals several key findings.

First, the SPS algorithm yielded an identical median score of 6 for both ICIs and co-medications, indicating that clinical assessment alone was often insufficient to distinguish the causative agent. This ambiguity highlights a significant diagnostic challenge, underscoring the need for more objective tools to guide clinical decisions in patients receiving multiple therapies.

To address this limitation, the LTT provided crucial in vitro evidence of drug-specific T-cell sensitization. Positive LTT results, with stimulation indices well above the established cut-off (3.9–13.6 for ICIs; and 2.2–25.8 for co-medications), allowed for a more definitive identification of the causative agent. This moved the assessment beyond clinical suspicion to an evidence-based conclusion, demonstrating the LTT’s value as a differentiating tool.

Furthermore, the data reveal a substantial risk associated with ICI rechallenge. Of the eight rechallenge events, five (45.5%) resulted in high-grade irAEs, primarily pneumonitis and hepatitis. This finding represents a clinically significant signal, suggesting that re-initiating ICI therapy after an irAE carries a high risk of inducing another severe, and potentially life-threatening, event. In contrast, patients who were not rechallenged based on strong LTT evidence of sensitization to a concomitant drug or excipient (e.g., dexketoprofen, hydrochlorothiazide, paclitaxel) showed no further immune-related complications, further validating this targeted diagnostic approach.

## 4. Discussion

This case series provides real-world evidence on the role of the LTT in determining the causality of irAEs caused by ICIs. It also explores its potential utility for informing rechallenge decisions. The findings extend prior meta-analyses and observational studies by incorporating immunological diagnostic testing with pharmacovigilance data, thereby bridging a translational gap between immunotoxicity detection and clinical decision-making.

Our findings indicate that the recurrence rate of irAEs following ICI rechallenge, 81.8% for all grades and 45.5% for high-grade events, was notably higher than that reported in the meta-analysis by Zhao et al. (2021) [11], in which the recurrence rate of all-grade irAEs following rechallenge was 34.2%, with high-grade recurrences occurring in 11.7% of cases. Similarly, Santini et al., 2018 [5] in a real-world cohort of 482 patients with advanced non–small cell lung cancer (NSCLC) treated with anti-PD-(L)1 agents, found that among 68 patients (14%) who developed serious irAEs requiring treatment interruption, 56% were successfully retreated, with 48% showing no recurrent toxicity, while 26% experienced recurrence and 26% developed new irAEs, most of which were mild and manageable. These findings support the feasibility of rechallenge but also underscore that outcomes are highly context-dependent. In a more recent systematic review stratified by cancer type and treatment, reported mean high-grade irAE rates were 19.7%, with variability across tumor and regimen (Jayathilaka et al., 2025) [6]. In contrast, our real-world case series identified considerably higher recurrence rates among rechallenged patients. The disparity between controlled datasets and our real-world experience likely reflects differences in patient selection and clinical complexity. Prospective trials and meta-analyses often exclude patients with multiple comorbidities, polypharmacy, or severe prior irAEs, while our case series embodies the heterogeneity of oncology practice, in which therapeutic decisions must balance efficacy with safety amidst diagnostic uncertainty.

Regarding organ specificity, our cohort aligns with published data, which show that hepatic, pulmonary, and renal involvement are the most frequent targets (Haanen et al., 2022) [1]. Notably, among rechallenged patients, pneumonitis emerged as the predominant recurrent irAE. This is consistent with observations from both Haanen et al. (2022) [1] and Jayathilaka et al. (2025) [6], who highlight respiratory and hepatic events as particularly prone to recurrence and severity upon rechallenge.

Although the present study was not designed to assess oncological outcomes, it is noteworthy that three of the eight rechallenged patients achieved disease stabilization, one developed pulmonary sequela, two experienced relapses of the underlying malignancy, and two died due to disease progression. Among these, IRAE10 exhibited a positive LTT result for ipilimumab (IPI) and was subsequently rechallenged with nivolumab (NIV), successfully completing oncological treatment without recurrence. These findings exemplify the clinical trade-off between maintaining antitumor control and avoiding potentially life-threatening toxicity. In line with previous findings by Zhao et al. (2021) [11], who reported an objective response rate (ORR) of 43.1% and a disease control rate (DCR) of 71.9% following ICI rechallenge, our results suggest that efficacy may be preserved, but only at the cost of increased risk in unselected, real-world populations.

A distinctive feature of this series of cases is the incorporation of immunological testing, specifically LTT, to assess T-cell sensitization to both ICIs and concomitant medications. Five patients showed markedly positive LTTs to ICIs, all of whom had experienced high-grade irAEs and were not rechallenged. The mechanistic basis of LTT in the setting of checkpoint blockade remains inherently limited, as ICIs induce systemic immune disinhibition rather than classic hapten-driven sensitization. However, the positive results observed in this study suggest that, for a subset of patients, irAEs may involve a focused expansion of drug-reactive memory T cells superimposed on the broader immune activation typical of ICI therapy. Furthermore, six patients exhibited sensitization to concomitant medications or excipients (e.g., polysorbate 80), and only three patients exhibited dual positivity. In these cases, avoidance of the offending non-ICI agent prevented further immune complications, thereby reinforcing the potential of the LTT to discriminate between ICI-induced and co-medication-related events.

These observations underscore several clinically significant points. Firstly, reliance solely on clinical algorithms, such as the Spanish Pharmacovigilance System (SPS) scale, may be insufficient for accurately identifying the causative drug in patients receiving multiple concurrent treatment therapies. The incorporation of in vitro assays, such as the LTT, could improve diagnostic precision and support personalized pharmacovigilance. Secondly, although the severity and type of the initial irAE influenced decisions regarding rechallenge, neither the timing of reintroduction nor the underlying malignancy predicted recurrence, aligning with previous findings (Zhao et al., 2021) [11]. This emphasizes the multifactorial nature of immune reactivation, which likely results from a combination of host predisposition, immune memory, and drug pharmacodynamics.

The LTT remains an investigational instrument with significant methodological limitations. False-negative results may occur due to transient loss of reactive lymphocytes after immunosuppression, and there is a lack of standardization across laboratories. Moreover, as the assay reflects peripheral rather than tissue-resident immune responses, its predictive capacity for organ-specific irAEs remains uncertain. Nonetheless, in the context of comprehensive causality assessment, it may provide valuable supportive evidence, particularly when clinical and pharmacological data are equivocal.

### 4.1. Strengths and Limitations

A primary strength of this research is reflected in its real-world, multidisciplinary design, through which clinical and immunological data were integrated to evaluate the utility of LTT in the assessment of irAEs. Clinically significant insights into safety management within high-risk settings were gained through the inclusion of patients who underwent ICI rechallenge. Furthermore, the interpretation of LTT results within structured causality algorithms was applied, providing a translational framework by which laboratory findings were linked to clinical pharmacovigilance, thereby fostering a more personalized understanding of immune toxicity.

Our study is subject to several limitations. The small sample size and retrospective design hinder the ability to draw statistical inferences. Subgroup analyses were not feasible due to the limited number of cases, and we note that larger, multi-center cohorts will be necessary to evaluate such differences reliably in the future; therefore, these findings should be considered as exploratory and hypothesis-generating. Additionally, a further limitation is the absence of a comparator group, which restricts the generalizability of our findings, given that this was an exploratory real-world series. A control group was not available, and the study does not aim to establish diagnostic accuracy.

Moreover, because the LTT was conducted after the resolution of acute events, the time interval between exposure and testing may have affected sensitivity. Unlike larger datasets that rely on standardized protocols, our approach integrates clinical reasoning with immunodiagnostic testing, providing a nuanced understanding of the evolving field of irAE management.

The external validity of these results is inherently constrained by the study design and sample size; however, the cohort reflects real-world clinical practice in a tertiary oncology setting, with patients treated using widely employed ICIs and managed according to current pharmacovigilance standards. Consequently, these findings may be relevant to comparable populations and serve as a foundation for future research aimed at validating the generalizability of LTT-based causality assessment in broader clinical contexts.

### 4.2. Clinical Implications and Future Directions

The LTT may serve as a valuable supplementary tool in the assessment of drug-induced and -related adverse events, particularly in cases where conventional diagnostic methods are either unsafe or produce inconclusive outcomes. By detecting drug-specific T-cell activation, LTT provides objective evidence that can enhance existing algorithms for causality assessment in pharmacovigilance and immunotoxicity.

A comprehensive immune profiling of PBMCs—such as the characterization of tumor-specific and non-tumor-specific immune subsets—was not available in this real-world cohort, as these analyses are not part of routine clinical practice. Future prospective studies integrating advanced immunophenotyping with LTT may help elucidate the immunological mechanisms underpinning irAEs and enhance clinical decision-making regarding ICI rechallenge.

## 5. Conclusions

We suggest that LTT and related immunodiagnostic assays could meaningfully enhance clinical evaluation, particularly for patients being considered for ICI rechallenge after experiencing high-grade toxicity. Prospective research is necessary to confirm their effectiveness in predicting safety and their role in guiding future rechallenge decisions, balancing antitumor efficacy with immune-related safety.

## Figures and Tables

**Figure 1 jcm-14-08596-f001:**
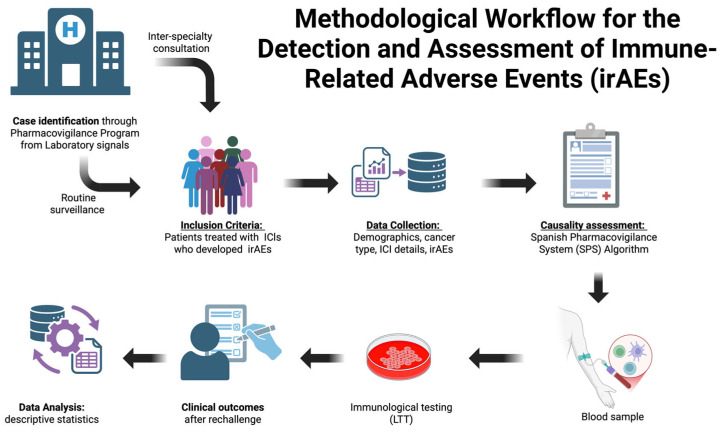
Shows how cases were identified through the institutional pharmacovigilance program. Created in https://BioRender.com (accessed on 23 November 2025).

**Figure 2 jcm-14-08596-f002:**
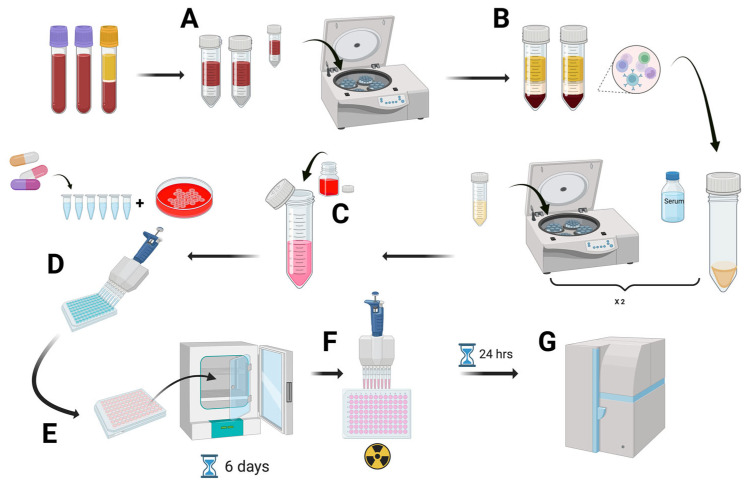
Schematic workflow of the lymphocyte transformation test (LTT)—Created in https://BioRender.com (accessed on 23 November 2025). (**A**) Peripheral blood is collected in heparinized tubes and centrifuged to separate plasma and the buffy coat. (**B**) PBMCs are isolated by Ficoll density gradient centrifugation. (**C**) Cells are resuspended in RPMI-1640 supplemented with HEPES and either AB serum (20%) or autologous plasma (10%). (**D**) Pure drug preparations are added in a dose–response format after toxicity testing. (**E**) PBMCs are plated in microtiter plates and incubated for 6 days at 37 °C, 5% CO_2_. (**F**) 3H-thymidine is added for 10–14 h to quantify proliferation. (**G**) Cells are harvested after 24 h, and β-emission is measured to determine drug-specific T-cell responses.

**Table 1 jcm-14-08596-t001:** Baseline characteristics of seventeen patients.

Case	Sex	Age	Diagnosis	ICI	Target	Time to Onset (Days)	Type of irAEs	Grade	Initial Grade
IRAE01	Male	58	Non-Small Cell Lung Carcinoma (NSCLC)	PEM	PD-1	104	ATIN	G1	Low grade
IRAE02	Female	63	Lung adenocarcinoma	PEM	PD-1	5	Pneumonitis	G3	High grade
IRAE03	Female	72	Large Cell Lung Carcinoma (LCLC)	PEM	PD-1	151	Macular rash	G1	Low grade
IRAE04	Male	76	Melanoma	NIV IPI	PD-1 CTLA-4	41	Hypophysitis	G1	Low grade
						118	Pneumonitis	G3	High grade
						41	Hepatitis	G1	Low grade
						41	Colitis	G1	Low grade
IRAE05	Male	74	Cutaneous Squamous Cell Carcinoma (cSCC)	PEM	PD-1	357	ATIN	G1	Low grade
IRAE06	Female	45	Medullary Thyroid Carcinoma (MTC)	DUR TREM	PD-L1	43	Meningoencephalitis	G4	High grade
							Arthritis	G1	Low grade
							Conjunctivitis	G1	Low grade
IRAE07	Male	62	Urothelial Carcinoma of the Bladder	DUR	PD-L1	118	Hepatitis	G3	High grade
IRAE08	Female	56	Melanoma	PEM	PD-1	497	Sensorimotor Axonal Neuropathy	G3	High grade
						136	Cutaneous Lupus	G1	Low grade
IRAE09	Male	76	Urothelial Carcinoma of the Bladder	ATEZ	PD-L1	25	Myositis	G2	Low grade
IRAE10	Male	56	Choroidal Melanoma of the Left Eye	NIV	PD-1	60	Hepatitis	G4	High grade
				IPI	CTLA-4	60	Hepatitis	G4	High grade
IRAE11	Male	70	Adenocarcinoma of the Cecum	PEM	PD-1	84	ATIN	G2	Low grade
IRAE12	Varón	62	Non-Small Cell Lung Carcinoma (NSCLC)	DUR	PD-L1	74	Pneumonitis	G4	High grade
IRAE013	Female	74	Lung adenocarcinoma	PEM	PD-1	13	ATIN	G2	Low grade
IRAE14	Male	67	Squamous Cell Carcinoma of the Lung	PEM	PD-1	231	Hepatitis	G3	High grade
IRAE15	Male	74	Squamous Cell Carcinoma of the Lung	NIV	PD-1	41	Hepatitis	G3	High grade
						41	Nephritis	G3	High grade
IRAE16	Male	61	Squamous Cell Carcinoma of the Lung	PEM	PD-1	123	Pneumonitis	G3	High grade
IRAE17	Male	55	Clear Cell Renal Cell Carcinoma (ccRCC)	NIV IPI	PD-1	246	Hepatitis	G3	High grade
							Pneumonitis	G3	High grade

ATEZ: Atezolizumab; CTCAE: Common Terminology Criteria for Adverse Events; CTLA-4: Cytotoxic T-lymphocyte–associated antigen 4; DUR: Durvalumab; G: Grade; indicates the severity of the adverse event according to CTCAE classification (G1: Mild, G2: Moderate, G3: Severe, G4: Life-threatening); ICI: Immune Checkpoint Inhibitor; IPI: Ipilimumab; irAE: Immune-Related Adverse Event; NIV: Nivolumab; PD-1: Programmed Death-1; PD-L1: Programmed Death-Ligand 1; PEM: Pembrolizumab; SPS: Spanish Pharmacovigilance System; TREM: Tremelimumab.

**Table 2 jcm-14-08596-t002:** Comprehensive Causality Assessment, Clinical Recommendations, and Final Patient Outcomes.

Case	ICI	Score SPS ICIs	LTT ICI (Cut-Off ≥ 2)	Concomitant Medication	Score SPS Other	LTT Other	Recommendations	Corticoid Therapy	Outcome	Comments
IRAE01	PEM	8	3.9	Dexketoprofen	8	1.1	Contraindication to PEM	High dose	No recurrence	Dexketoprofen negative
				Omeprazole	8	1.4				
IRAE02	PEM	6	2				Continue treatment	High dose	No recurrence	Tolerated pembrolizumab following the adverse event
IRAE03	PEM	5	1.6	Dexamethasone	4	1.5	Continue treatment	Moderate dose	Death	Tolerated pembrolizumab following the adverse event
				Dextromethorphan	4	1.8				
				Tinzaparin	5	1.8				
IRAE04	NIV	3	1.5	Enalapril	8	25.8	None	High dose	Death	
	IPI	8	10.7				Contraindication to IPI	High dose	Death	
IRAE05	PEM	5	1.1	Esomeprazol	5	0.9	None	No	No recurrence	Tolerated pembrolizumab following the adverse event
				Metamizol	5	1.6				
				Furosemide	5	1.7				
IRAE06	DUR TREM	6	1.6	Tremelimumab	8	NH	Contraindication to DUR and TREM	Moderate dose	No recurrence	
IRAE07	DUR	6	13.6	Enfortumab vedotina	4	3.4	Avoid (risk/benefit assessment) DUR y Enfortumab vedotina	High dose	No recurrence	
IRAE08	PEM	3	1.6	Rosuvastatin	6	2.2	Contraindication to statins, ICI therapy can induce sensitization	High dose	Recurrence	
IRAE09	ATEZ	5	0.9	Pitavastatin	6	1.1	Contraindication to Pivastatin and statins. Check CPK while treatment with atezolizumab.	High dose	Death	
IRAE10	NIV	7	1.5	Dexketoprofen	4	1.2	Contraindication to IPI	High dose	No recurrence	Tolerated nivolumab following the adverse event
	IPI	7	1.2	Metamizol	4	1.2				
				Paracetamol	4	1				
				Clindamycin	3	1.4				
IRAE11	PEM	5	2.3	Ibuprophen	8	6.7	Check renal function while treatment with PEM.	High dose	No recurrence	Tolerated pembrolizumab following the adverse event
IRAE12	DUR	8	4.2	Carboplatin	6	0.7	Avoidance of carboplatin, pemetrexed, and durvalumab is recommended. There is no contraindication for the remaining ICIs.	High dose	Death	
				Pemetrexed	6	1.7				
IRAE013	PEM	7	0.8	Dexketoprofen	8	4.9	Contraindications to dexketoprofen and other NSAIDs of the arylpropionic acid class, thiazide diuretics, and pembrolizumab.	High dose	Recurrence	Patch test to hydrochlorothiazide, dexketoprofen, and omeprazole: negative
				Hydrochlorothiazide	8	3.2				
				Metamizol	4	2				
				Omeprazol	4	1.2				
IRAE14	PEM	4	1.6	Pyrazinamide	9	1.3	Monitor hepatic function throughout the course of pembrolizumab treatment	No dose	Death	Positive Flow Cytometry to Pyrazinamide
				Isoniazid	9	1.5				
				Rifampicin	9	1				
IRAE15	NIV	5	10.8	Polisorbate 80		4.2	G3 irAE: mixed hepatitis G3, due to sensitization to excipients alcohols of the polysorbate 80 type; patient was taking omeprazole, atorvastatin 80 mg, and NIV 17/03/24: Ps. 80 negative	High dose	No recurrence	Positive LTT to polysorbate 80. Grade 3 irAE/Grade 3 mixed hepatitis due to sensitization to excipients—alcohol derivatives such as polysorbate 80.
IRAE16	PEM	8	1	Paclitaxel	8		Contraindication to PEM	High dose	Recurrence	LTT with taxanes is not assessable, but flow cytometry with paclitaxel is positive.
IRAE17	NIV	4	0.8	Manidipine	4	1.1	Contraindication to NIV	High dose	Recurrence	
				Levofloxacin	5	0.8				
				Amoxicillin	5	0.8				

ATEZ: Atezolizumab; DUR: Durvalumab; ICI: Immune Checkpoint Inhibitor; irAE: Immune-Related Adverse Event; IPI: Ipilimumab; NIV: Nivolumab; LTT: Lymphocyte Transformation Test; an in vitro assay to detect drug-specific T-cell sensitization. The result is expressed as a stimulation index, with a value ≥ 2 considered positive; NH: Not Performed; NSAIDs: Nonsteroidal Anti-Inflammatory Drugs; PEM: Pembrolizumab; SPS: Spanish Pharmacovigilance System; a drug causality algorithm used to assess the likelihood that a drug caused an adverse event; TREM: Tremelimumab.

**Table 3 jcm-14-08596-t003:** Clinical Characteristics and Outcomes of Immune Checkpoint Inhibitor Rechallenge.

Initial Exposure and irAE						First Rechallenge and Outcome					Second Rechallenge and Outcome				
CASE	ICI	Time to Onset (Days)	Type of irAEs	Grade	Initial Grade	Rechallenge	ICIs Rechallenge	Time to Onset (Days)	Type of irAEs	Second Grade	Rechallenge	ICIs Rechallenge	Time to Onset (Days)	Type of irAEs	Third Grade
IRAE01	PEM	104	ATIN	G1	Low grade	No									
IRAE02	PEM	5	Pneumonitis	G3	High grade	Yes	PEM	17	Pneumonitis	High grade	Yes	PEM	5	Pneumonitis	High grade
IRAE03	PEM	151	Macular rash	G1	Low grade	Yes	PEM	84	Arthralgias	Low grade					
IRAE04	NIV	41	Hypophysitis	G1	Low grade	No									
	IPI	118	Pneumonitis	G3	High grade	No									
		41	Hepatitis	G1	Low grade										
		41	Colitis	G1	Low grade										
IRAE05	PEM	357	ATIN	G1	Low grade	Yes	PEM	902	Arthralgias	Low grade					
IRAE06	DUR	43	Meningoencephalitis	G4	High grade										
	TREM		Arthritis	G1	Low grade										
			Conjuntivitis	G1	Low grade										
IRAE07	DUR	118	Hepatitis	G3	High grade	No									
IRAE08	PEM	497	Sensorimotor Axonal Neuropathy	G3	High grade	No									
		136	Cutaneous Lupus	G1	Low grade										
IRAE09	ATEZ	25	Miositis	G2	Low grade	Yes	ATEZ	21	Myalgias	Low grade					
IRAE10	NIV	60	Hepatitis	G4	High grade	Yes	NIV	6	Hepatitis	High grade	Yes	NIV	No	No	
	IPI	60	Hepatitis	G4	High grade										
IRAE11	PEM	84	Tubulointerstitial nephritis	G2	Low grade	Yes	PEM	63	ATIN	Low grade	Yes	PEM	No	No	
IRAE12	DUR	74	Pneumonitis	G4	High grade	No									
IRAE013	PEM	13	ATIN	G2	Low grade	No									
IRAE14	PEM	231	Hepatitis	G3	High grade	No									
IRA15	NIV	41	Hepatitis	G3	High grade	No									
		41	Nephritis G3	G3	High grade										
IRA16	PEM	123	Pneumonitis	G3	High grade	Yes	PEM	30	Pneumonitis	High grade					
IRA17	NIV	246	Hepatitis	G3	High grade	Yes	NIV	126	Pneumonitis	High grade					
	IPI	246	Pneumonitis	G3	High grade										

ATEZ: Atezolizumab; ATIN: Acute Tubular Interstitial Nephritis; DUR: Durvalumab; ICI: Immune Checkpoint Inhibitor; irAE: Immune-Related Adverse Event; IPI: Ipilimumab; NIV: Nivolumab; PEM: Pembrolizumab; TREM: Tremelimumab.

**Table 4 jcm-14-08596-t004:** Summary of Causality Assessment and Rechallenge Outcomes.

Causality Assessment		
	Implicated ICIs	Co-medication
SPS Algorithm Score (range)	6 (3–8)	6 (3–9)
LTT SI ^a^ (Range)	3.9–13.6 ^a^	2.2–25.8
**Rechallenge Outcomes**		
High-Grade irAEs (*n* = 5)	*Pneumonitis* (*n* = 4)	45.5%
	*Hepatitis* (*n* = 1)	
Low-Grade irAEs (*n* = 4)	*Nephritis* (*n* = 1)	36.40%
	*Myositis* (*n* = 1)	
	*Arthralgia* (*n* = 2)	
No irAESs (*n* = 2)	*None* (*n* = 2)	18.20%
Total	11	100%

(^a^) cut-off ≥ 2; ICI: Immune Checkpoint Inhibitor; LTT: Lymphocyte Transformation Test; SI: Stimulation Index; SPS: Spanish Pharmacovigilance System.

## Data Availability

The datasets generated during and/or analyzed during the current study are available from the corresponding author on reasonable request.

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
