# Peer review of "The Role of the Lymphocyte Transformation Test in Immune-Related Adverse Events from Immune Checkpoint Inhibitors: A Case Series"

_jcm, 2025, doi:10.3390/jcm14238596_

Round 1
Reviewer 1 Report
Comments and Suggestions for Authors
This manuscript presents a small but focused case series exploring the role of the lymphocyte transformation test (LTT) in assessing immune-related adverse events (irAEs) in patients undergoing immune checkpoint inhibitor (ICI) therapy. The topic is clinically relevant given the rising use of ICIs and the complexity of managing irAEs. The authors attempt to correlate LTT results with clinical features and management outcomes, which may offer a potentially useful diagnostic adjunct.
However, there are limitations:
- The study includes only 5 cases, limiting generalizability and statistical rigor.
- The mechanistic rationale and limitations of the LTT in the context of ICI-induced irAEs could be better contextualized.
- No control group is included, and sensitivity/specificity of the test in this setting remains speculative.
- Figures and tables are helpful, but text flow could be improved for clarity.
- The English language quality is mostly acceptable but would benefit from light polishing.
Overall, the manuscript addresses a niche but emerging diagnostic question in oncology and immunotoxicology.
Author Response
RESPONSE TO REVIEWER 1
Comment 1: The study includes only 5 cases, limiting generalizability and statistical rigor.
Response
We thank the reviewer for this comment. The rarity and clinical heterogeneity of ICI-induced irAEs for which an LTT is clinically indicated make assembling larger cohorts challenging at present. We have now explicitly acknowledged this limitation in the revised manuscript, emphasizing that the small sample size precludes statistical generalization and that the study should be interpreted as exploratory and hypothesis-generating.
We have clarified this point in the revised manuscript in lines 395 – 398
Comment 2: The mechanistic rationale and limitations of the LTT in the context of ICI-induced irAEs could be better contextualized.
Response
We thank the reviewer for this insightful comment. We have now expanded the mechanistic contextualization of the LTT in the revised manuscript.
The revised paragraph appears in the Discussion section between lines 357 to 364
Comment 3: No control group is included, and sensitivity/specificity of the test in this setting remains speculative.
Response
We agree with the reviewer, as this was a retrospective case series of clinically indicated LTTs, and given that this was an exploratory real-world series. A control group was not available, and the study does not aim to establish diagnostic accuracy.
We have clarified this point in the revised manuscript in lines 398 - 400
Comment 4: Figures and tables are helpful, but text flow could be improved for clarity.
Response
We appreciate the reviewer's insightful observation. The Results section has been systematically reorganized to enhance the narrative flow, eliminate redundancy, and align figures more accurately with their respective descriptions.
The section has been subdivided into subsections in accordance with the methodology, with paragraphs revised from line 196 to 276.
Comment 5: The English language quality is mostly acceptable but would benefit from light polishing.
Response
We appreciate this comment. The manuscript has undergone a comprehensive language revision to improve clarity, grammar, and overall readability.
Comment 6: Overall, the manuscript addresses a niche but emerging diagnostic question in oncology and immunotoxicology.
Response
We thank the reviewer for this positive assessment. We believe the revised version further strengthens the clinical and methodological relevance of our work in this emerging area.
Sincerely,
Dr. Dueñas
Dr. González-Muñoz
Dr. Ramírez
(On behalf of all authors)

Reviewer 2 Report
Comments and Suggestions for Authors
Lopez et al in this well written manuscript describes the use of PBMC from patients treated with ICI therapy who had irAEs during treatment and after rechallenging and used these PBMC for LTT in vitro assay. The authors hope to use LTT to support clinical decision making for safety management of patients from a retrospective clinical trial to improve clinicians’ decision making. To enhance the manuscript the authors should make the following revisions.
- How are the immune profiles of these patient’s PBMCs (non-tumor/tumor specific T-cells and non-tumor/tumor specific immune cells)? Immune profiles should be compared to the LTT data to see if this can improve decision making for clinicians to predict if ICI therapy rechallenge could be safely given.
- A figure describing how the LTT assay is performed must be shown.
- It should be discussed that in an ideal situation and future studies that to further validate LTT data from PBMC’s, patient PBMC’s should be taken before and after treatment initiation of immune checkpoint therapy and/or treatments that are used to manage irAEs to see if it can predict irAEs and/of if treatment should be given again eventhough it was discontinued due to irAEs.
- Line 147-150 how close where these drug concentrations to the clinical settings (concentration found in systemic circulation)? This must be noted to ensure clinical translatability.
- Explain why Phytohemagglutinin is a positive control. What was the negative control used besides healthy controls (blank/vehicle)?
- Were there differences between sexes, and/or across treatment modalities (ant-CTLA-4/anti-PD-1/anti-PD-L1) for LTT? This reviewer acknowledges that this will be a small sample size.
- Table 2 should say yes instead of si.
Author Response
RESPONSE TO REVIEWER 2
Comment 1: How are the immune profiles of these patient’s PBMCs (non-tumor/tumor specific T-cells and non-tumor/tumor specificimmune cells)? Immune profiles should be compared to the LTT data to see if this can improve decision making for clinicians to predict if ICI therapy rechallenge could be safely given.
Response
We thank the reviewer for this important suggestion. However, a detailed immune profiling of PBMCs — including the distinction between tumor-specific and non–tumor-specific T cells or other immune cell subsets — is not performed in routine clinical practice for patients receiving ICIs. As this was a retrospective real-world case series, no such samples were collected, and therefore these data are not available for comparison with the LTT results.
We fully agree that integrating comprehensive immune profiling with LTT findings could provide valuable mechanistic insight and potentially improve clinical decision-making for ICI rechallenge. We have added this point to the Discussion as a relevant future research direction, highlighting the need for prospective studies combining functional immune analyses with LTT in the context of irAEs.
We have added a paragraph in the Discussion section between lines 418 to 423
Comment 2: A figure describing how the LTT assay is performed must be shown.
Response
We thank the reviewer for this important suggestion.
We have added an additional Figure 2 in the Methodology section
Figure 2. Schematic workflow of the lymphocyte transformation test (LTT) - Created with BioRender.com.
(A) Peripheral blood is collected in heparinized tubes and centrifuged to separate plasma and the buffy coat. (B) PBMCs are isolated by Ficoll density gradient centrifugation. (C) Cells are resuspended in RPMI-1640 supplemented with HEPES and either AB serum (20%) or autologous plasma (10%). (D) Pure drug preparations are added in a dose-response format after toxicity testing. (E) PBMCs are plated in microtiter plates and incubated for 6 days at 37 °C, 5% CO₂. (F) 3H-thymidine is added for 10–14 h to quantify proliferation. (G) Cells are harvested after 24 h, and β-emission is measured to determine drug-specific T-cell responses.
Comment 3: It should be discussed that in an ideal situation and future studies that to further validate LTT data from PBMC’s, patient PBMC’s should be taken before and after treatment initiation of immune checkpoint therapy and/or treatments that are used to manage irAEs to see if it can predict irAEs and/of if treatment should be given again even though it was discontinued due to irAEs.
Response
We thank the reviewer for this suggestion. Serial sampling could help improve the predictive value of LTT and help determine whether ICIs can be safely reintroduced after discontinuation due to toxicity. However, in our real-world retrospective cohort, PBMCs were not collected longitudinally, reflecting current clinical practice where routine immune profiling is not standard.
Comment 4: Line 147-150 how close where these drug concentrations to the clinical settings (concentration found in systemic circulation)? This must be noted to ensure clinical translatability
Response
We thank the reviewer for this observation. In our LTT protocol, the drug concentration curves include the clinically relevant plasma concentrations of the tested agents. Specifically, the range of concentrations used in vitro encompasses the therapeutic systemic levels reported for these drugs, ensuring that the assay conditions reflect exposure levels achievable in patients. We have now clarified this point in the Methods section to strengthen the clinical translatability of our LTT results.
We have added a text in the in the Methodology section between lines 164 to 166
Comment 5: Explain why Phytohemagglutinin is a positive control. What was the negative control used besides healthy controls(blank/vehicle)?
Response
We thank the reviewer for this question. Phytohemagglutinin (PHA) is used as a positive control in the LTT because it is a well-established, non–specific T-cell mitogen that reliably induces robust lymphocyte proliferation. Its inclusion confirms the viability and proliferative capacity of PBMCs and ensures that a lack of response to a tested drug is not due to impaired cell function.
Regarding the negative control, the assay included unstimulated PBMCs cultured with medium alone, which served as the baseline reference for calculating stimulation indices. Healthy donor controls were used in addition to this internal negative control, but the primary negative control within each assay run was the blank (medium-only) condition
Comment 6: Were there differences between sexes, and/or across treatment modalities (ant-CTLA-4/anti-PD-1/anti-PD-L1) for LTT? This reviewer acknowledges that this will be a small sample size.
Response
We thank the reviewer for this comment. However, given the very small sample size, the study is underpowered to detect meaningful differences by sex or across treatment modalities (anti-CTLA-4, anti-PD-1, anti-PD-L1). Although we examined the dataset for potential qualitative patterns, no consistent or interpretable trends emerged.
We have now clarified in the revised manuscript that subgroup analyses were not feasible due to the limited number of cases, and we note that larger, multi-centre cohorts will be necessary to evaluate such differences reliably in the future.
We have clarified this point in the revised manuscript in lines 395 – 398
Comment 7: Table 2 should say yes instead of si.
Response
We appreciate the reviewer's attention to this typographical error. The correction has been made by changing “si” to “yes” in Table 2, which now corresponds to Table 3 in the revised manuscript.
We again thank both reviewers for their constructive comments, which have significantly improved the quality and clarity of our manuscript. We hope that the revised version satisfactorily addresses all concerns.
Sincerely,
Dr. Dueñas
Dr. González-Muñoz
Dr. Ramírez
(On behalf of all authors)

Round 2
Reviewer 2 Report
Comments and Suggestions for Authors
Thank you for addressing my comments. Excellent job.